# Expectation Backpropagation: Parameter-Free Training of Multilayer Neural Networks with Continuous or Discrete Weights

**Daniel Soudry**[1], **Itay Hubara**[2], **Ron Meir**[2]
(1) Department of Statistics, Columbia University
(2) Department of Electrical Engineering, Technion, Israel Institute of Technology
`daniel.soudry@gmail.com,itayhubara@gmail.com,rmeir@ee.technion.ac.il`

## Abstract

Multilayer Neural Networks (MNNs) are commonly trained using gradient descent-based methods, such as BackPropagation (BP). Inference in probabilistic graphical models is often done using variational Bayes methods, such as Expectation Propagation (EP). We show how an EP based approach can also be used to train deterministic MNNs. Specifically, we approximate the posterior of the weights given the data using a "mean-field" factorized distribution, in an online setting. Using online EP and the central limit theorem we find an analytical approximation to the Bayes update of this posterior, as well as the resulting Bayes estimates of the weights and outputs.

Despite a different origin, the resulting algorithm, Expectation BackPropagation (EBP), is very similar to BP in form and efficiency. However, it has several additional advantages: (1) Training is parameter-free, given initial conditions (prior) and the MNN architecture. This is useful for large-scale problems, where parameter tuning is a major challenge. (2) The weights can be restricted to have discrete values. This is especially useful for implementing trained MNNs in precision limited hardware chips, thus improving their speed and energy efficiency by several orders of magnitude.

We test the EBP algorithm numerically in eight binary text classification tasks. In all tasks, EBP outperforms: (1) standard BP with the optimal constant learning rate (2) previously reported state of the art. Interestingly, EBP-trained MNNs with binary weights usually perform better than MNNs with continuous (real) weights - if we average the MNN output using the inferred posterior.

## 1 Introduction

Recently, Multilayer[1] Neural Networks (MNNs) with deep architecture have achieved state-of-the-art performance in various supervised learning tasks [11, 14, 8]. Such networks are often massive and require large computational and energetic resources. A dense, fast and energetically efficient hardware implementation of trained MNNs could be built if the weights were restricted to discrete values. For example, with binary weights, the chip in [13] can perform $10^{12}$ operations per second with 1mW power efficiency. Such performances will enable the integration of massive MNNs into small and low-power electronic devices.

Traditionally, MNNs are trained by minimizing some error function using BackPropagation (BP) or related gradient descent methods [15]. However, such an approach cannot be directly applied if the weights are restricted to binary values. Moreover, crude discretization of the weights is usually quite

destructive [20]. Other methods have been suggested in the 90's (*e.g.*, [23, 3, 18]), but it is not clear whether these approaches are scalable.

The most efficient methods developed for training Single-layer[2] Neural Networks (SNN) with binary weights use approximate Bayesian inference, either implicitly [6, 1] or explicitly [24, 22]. In theory, given a prior, the Bayes estimate of the weights can be found from their posterior given the data. However, storing or updating the full posterior is usually intractable. To circumvent this problem, these previous works used a factorized "mean-field" form the posterior of the weights given the data.

As explained in [22], this was done using a special case of the widely applicable Expectation Propagation (EP) algorithm [19] - with an additional approximation that the fan-in of all neurons is large, so their inputs are approximately Gaussian. Thus, given an error function, one can analytically obtain the Bayes estimate of the weights or the outputs, using the factorized approximation of the posterior. However, to the best of our knowledge, it is still unknown whether such an approach could be generalized to MNNs, which are more relevant for practical applications.

In this work we derive such generalization, using similar approximations (section 3). The end result is the Expectation BackPropagation (EBP, section 4) algorithm for online training of MNNs where the weight values can be either continuous (*i.e.*, real numbers) or discrete (*e.g.*, $\pm 1$ binary). Notably, the training is parameter-free (with no learning rate), and insensitive to the magnitude of the input. This algorithm is very similar to BP. Like BP, it is very efficient in each update, having a linear computational complexity in the number of weights.

We test the EBP algorithm (section 5) on various supervised learning tasks: eight high dimensional tasks of classifying text into one of two semantic classes, and one low dimensional medical discrimination task. Using MNNs with two or three weight layers, EBP outperforms both standard BP, as well as the previously reported state of the art for these tasks [7]. Interestingly, the best performance of EBP is usually achieved using the Bayes estimate of the output of MNNs with *binary* weights. This estimate can be calculated analytically, or by averaging the output of several such MNNs, with weights sampled from the inferred posterior.

## 2 Preliminaries

**General Notation**  A non-capital boldfaced letter $\mathbf{x}$ denotes a column vector with components $x_i$, a boldfaced capital letter $\mathbf{X}$ denotes a matrix with components $X_{ij}$. Also, if indexed, the components of $\mathbf{x}_l$ are denoted $x_{i,l}$ and those of $\mathbf{X}_l$ are denoted $X_{ij,l}$. We denote by $P(x)$ the probability distribution (in the discrete case) or density (in the continuous case) of a random variable $X$, $P(x|y) = P(x,y)/P(y), \langle x \rangle = \int x P(x)\, dx, \langle x|y \rangle = \int x P(x|y)\, dx, \mathrm{Cov}(x,y) = \langle xy \rangle - \langle x \rangle \langle y \rangle$ and $\mathrm{Var}(x) = \mathrm{Cov}(x,x)$. Integration is exchanged with summation in the discrete case. For any condition $\mathcal{A}$, we make use of $\mathcal{I}\{A\}$, the indicator function (*i.e.*, $\mathcal{I}\{A\} = 1$ if $A$ holds, and zero otherwise), and $\delta_{ij} = \mathcal{I}\{i = j\}$, Kronecker's delta function. If $\mathbf{x} \sim \mathcal{N}(\boldsymbol{\mu}, \boldsymbol{\Sigma})$ then it is Gaussian with mean $\boldsymbol{\mu}$ and covariance matrix $\boldsymbol{\Sigma}$, and we denote its density by $\mathcal{N}(\mathbf{x}|\boldsymbol{\mu}, \boldsymbol{\Sigma})$. Furthermore, we use the cumulative distribution function $\Phi(x) = \int_{-\infty}^{x} \mathcal{N}(u|0,1)\, du$.

**Model**  We consider a general feedforward Multilayer Neural Network (MNN) with connections between adjacent layers (Fig. 2.1). For analytical simplicity, we focus here on deterministic binary ($\pm 1$) neurons. However, the framework can be straightforwardly extended to other types of neurons (deterministic or stochastic). The MNN has $L$ layers, where $V_l$ is the width of the $l$-th layer, and $\mathcal{W} = \{\mathbf{W}_l\}_{l=1}^{L}$ is the collection of $V_l \times V_{l-1}$ synaptic weight matrices which connect neuronal layers sequentially. The outputs of the layers are $\{\mathbf{v}_l\}_{l=0}^{L}$, where $\mathbf{v}_0$ is the input layer, $\{\mathbf{v}_l\}_{l=1}^{L-1}$ are the hidden layers and $\mathbf{v}_L$ is the output layer. In each layer,

$$\mathbf{v}_l = \mathrm{sign}(\mathbf{W}_l \mathbf{v}_{l-1}) \tag{2.1}$$

where each sign "activation function" (a neuronal layer) operates component-wise (*i.e.*, $\forall i : (\mathrm{sign}(\mathbf{x}))_i = \mathrm{sign}(x_i)$). The output of the network is therefore

$$\mathbf{v}_L = g(\mathbf{v}_0, \mathcal{W}) = \mathrm{sign}(\mathbf{W}_L \mathrm{sign}(\mathbf{W}_{L-1} \mathrm{sign}(\cdots \mathbf{W}_1 \mathbf{v}_0))) . \tag{2.2}$$

We assume that the weights are constrained to some set $\mathcal{S}$, with the specific restrictions on each weight denoted by $S_{ij,l}$, so $W_{ij,l} \in S_{ij,l}$ and $\mathcal{W} \in \mathcal{S}$. If $S_{ij,l} = \{0\}$, then we say that $W_{ij,l}$ is "disconnected". For simplicity, we assume that in each layer the "fan-in" $K_l = |\{j|S_{ij,l} \neq \{0\}\}|$ is constant for all $i$. Biases can be optionally included in the standard way, by adding a constant output $v_{0,l} = 1$ to each layer.

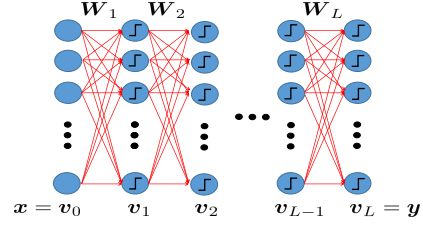

Figure 2.1: Our MNN model (Eq. 2.2).

**Task** We examine a supervised classification learning task, in Bayesian framework. We are given a *fixed* set of sequentially labeled data pairs $D_N = \left\{ \mathbf{x}^{(n)}, \mathbf{y}^{(n)} \right\}_{n=1}^{N}$ (so $D_0 = \emptyset$), where each $\mathbf{x}^{(n)} \in \mathbb{R}^{V_0}$ is a data point, and each $\mathbf{y}^{(n)}$ is a label taken from a binary set $\mathcal{Y} \subset \{-1, +1\}^{V_L}$. For brevity, we will sometimes suppress the sample index $n$, where it is clear from the context. As common for supervised learning with MNNs, we assume that for all $n$ the relation $\mathbf{x}^{(n)} \to \mathbf{y}^{(n)}$ can be represented by a MNN with known architecture (the 'hypothesis class'), and unknown weights $\mathcal{W} \in \mathcal{S}$. This is a reasonable assumption since a MNN can approximate any deterministic function, given that it has sufficient number of neurons [12] (if $L \geq 2$). Specifically, there exists some $\mathcal{W}^* \in \mathcal{S}$, so that $\mathbf{y}^{(n)} = f\left(\mathbf{x}^{(n)}, \mathcal{W}^*\right)$ (see Eq. 2.2). Our goals are: (1) estimate the most probable $\mathcal{W}^*$ for this MNN, (2) estimate the most probable $\mathbf{y}$ given some (possibly unseen) $\mathbf{x}$.

# 3 Theory

In this section we explain how a specific learning algorithm for MNNs (described in section 4) arises from approximate (mean-field) Bayesian inference, used in this context (described in section 2).

## 3.1 Online Bayesian learning in MNNs

We approach this task within a Bayesian framework, where we assume some prior distribution on the weights - $P\left(\mathcal{W}|D_0\right)$. Our aim is to find $P\left(\mathcal{W}|D_N\right)$, the posterior probability for the configuration of the weights $\mathcal{W}$, given the data. With this posterior, one can select the most probable weight configuration - the Maximum A Posteriori (MAP) weight estimate

$$\mathcal{W}^* = \operatorname{argmax}_{\mathcal{W} \in \mathcal{S}} P\left(\mathcal{W}|D_N\right), \tag{3.1}$$

minimizing the expected zero-one loss *over the weights* ($\mathcal{I}\left\{\mathcal{W}^* \neq \mathcal{W}\right\}$). This weight estimate can be implemented in a single MNN, which can provide an estimate of the label $\mathbf{y}$ for (possibly unseen) data points $\mathbf{x}$ through $\mathbf{y} = g\left(\mathbf{x}, \mathcal{W}^*\right)$. Alternatively, one can aim to minimize the expected loss *over the output* - as more commonly done in the MNN literature. For example, if the aim is to reduce classification error then one should use the MAP output estimate

$$\mathbf{y}^* = \operatorname{argmax}_{\mathbf{y} \in \mathcal{Y}} \sum_{\mathcal{W}} \mathcal{I}\left\{g\left(\mathbf{x}, \mathcal{W}\right) = \mathbf{y}\right\} P\left(\mathcal{W}|D_N\right), \tag{3.2}$$

which minimizes the zero-one loss ($\mathcal{I}\left\{\mathbf{y}^* \neq g\left(\mathbf{x}, \mathcal{W}\right)\right\}$) over the outputs. The resulting estimator does not generally have the form of a MNN (*i.e.*, $\mathbf{y} = g\left(\mathbf{x}, \mathcal{W}\right)$ with $\mathcal{W} \in \mathcal{S}$), but can be approximated by averaging the output over many such MNNs with $\mathcal{W}$ values sampled from the posterior. Note that averaging the output of several MNNs is a common method to improve performance.

We aim to find the posterior $P\left(\mathcal{W}|D_N\right)$ in an online setting, where samples arrive sequentially. After the $n$-th sample is received, the posterior is updated according to Bayes rule:

$$P\left(\mathcal{W}|D_n\right) \propto P\left(\mathbf{y}^{(n)}|\mathbf{x}^{(n)}, \mathcal{W}\right) P\left(\mathcal{W}|D_{n-1}\right), \tag{3.3}$$

for $n = 1, \ldots, N$. Note that the MNN is *deterministic*, so the likelihood (per data point) has the following simple form[3]

$$P\left(\mathbf{y}^{(n)}|\mathbf{x}^{(n)}, \mathcal{W}\right) = \mathcal{I}\left\{g\left(\mathbf{x}^{(n)}, \mathcal{W}\right) = \mathbf{y}^{(n)}\right\}. \tag{3.4}$$

Therefore, the Bayes update in Eq. 3.3 simply makes sure that $P\left(\mathcal{W}|D_n\right) = 0$ in any "illegal" configuration (*i.e.*, any $\mathcal{W}^0$ such that $g\left(\mathbf{x}^{(k)}, \mathcal{W}^0\right) \neq \mathbf{y}^{(k)}$ for some $1 \leq k \leq n$. In other words, the posterior is equal to the prior, restricted to the "legal" weight domain, and re-normalized appropriately. Unfortunately, this update is generally intractable for large networks, mainly because we need to store and update an exponential number of values for $P\left(\mathcal{W}|D_n\right)$. Therefore, some approximation is required.

## 3.2   Approximation 1: mean-field

In order to reduce computational complexity, instead of storing $P\left(\mathcal{W}|D_n\right)$, we will store its factorized ('mean-field') approximation $\hat{P}\left(\mathcal{W}|D_n\right)$, for which

$$\hat{P}\left(\mathcal{W}|D_n\right) = \prod_{i,j,l} \hat{P}\left(W_{ij,l}|D_n\right) , \tag{3.5}$$

where each factor must be normalized. Notably, it is easy to find the MAP estimate of the weights (Eq. 3.1) under this factorized approximation $\forall i,j,l$

$$W_{ij,l}^* = \mathrm{argmax}_{W_{ij,l} \in S_{ij,l}} \hat{P}\left(W_{ij,l}|D_N\right) . \tag{3.6}$$

The factors $\hat{P}\left(W_{ij,l}|D_n\right)$ can be found using a standard variational approach [5, 24]. For each $n$, we first perform the Bayes update in Eq. 3.3 with $\hat{P}\left(\mathcal{W}|D_{n-1}\right)$ instead of $P\left(\mathcal{W}|D_{n-1}\right)$. Then, we project the resulting posterior onto the family of distributions factorized as in Eq. 3.5, by minimizing the *reverse* Kullback-Leibler divergence (similarly to EP [19, 22]). A straightforward calculation shows that the optimal factor is just a marginal of the posterior (appendix A, available in the supplementary material). Performing this marginalization on the Bayes update and re-arranging terms, we obtain a Bayes-like update to the marginals $\forall i,j,l$

$$\hat{P}\left(W_{ij,l}|D_n\right) \propto \hat{P}\left(\mathbf{y}^{(n)}|\mathbf{x}^{(n)}, W_{ij,l}, D_{n-1}\right) \hat{P}\left(W_{ij,l}|D_{n-1}\right) , \tag{3.7}$$

where

$$\hat{P}\left(\mathbf{y}^{(n)}|\mathbf{x}^{(n)}, W_{ij,l}, D_{n-1}\right) = \sum_{\mathcal{W}':W'_{ij,l}=W_{ij,l}} P\left(\mathbf{y}^{(n)}|\mathbf{x}^{(n)}, \mathcal{W}'\right) \prod_{\{k,r,m\}\neq\{i,j,l\}} \hat{P}\left(W'_{kr,m}|D_{n-1}\right) \tag{3.8}$$

is the marginal likelihood. Thus we can directly update the factor $\hat{P}\left(W_{ij,l}|D_n\right)$ in a single step. However, the last equation is still problematic, since it contains a generally intractable summation over an exponential number of values, and therefore requires simplification. For simplicity, from now on we replace any $\hat{P}$ with $P$, in a slight abuse of notation (keeping in mind that the distributions are approximated).

## 3.3   Simplifying the marginal likelihood

In order to be able to use the update rule in Eq. 3.7, we must first calculate the marginal likelihood $P\left(\mathbf{y}^{(n)}|\mathbf{x}^{(n)}, W_{ij,l}, D_{n-1}\right)$ using Eq. 3.8. For brevity, we suppress the index $n$ and the dependence on $D_{n-1}$ and $\mathbf{x}$, obtaining

$$P\left(\mathbf{y}|W_{ij,l}\right) = \sum_{\mathcal{W}':W'_{ij,l}=W_{ij,l}} P\left(\mathbf{y}|\mathcal{W}'\right) \prod_{\{k,r,m\}\neq\{i,j,l\}} P\left(W'_{kr,m}\right) , \tag{3.9}$$

where we recall that $P\left(\mathbf{y}|\mathcal{W}'\right)$ is simply an indicator function (Eq. 3.4). Since, by assumption, $P\left(\mathbf{y}|\mathcal{W}'\right)$ arises from a feed-forward MNN with input $\mathbf{v}_0 = \mathbf{x}$ and output $\mathbf{v}_L = \mathbf{y}$, we can perform the summations in Eq. 3.9 in a more convenient way - layer by layer. To do this, we define

$$P\left(\mathbf{v}_m|\mathbf{v}_{m-1}\right) = \sum_{\mathbf{W}'_m} \prod_{k=1}^{V_m} \left[ \mathcal{I}\left\{ v_{k,m} \sum_{r=1}^{V_{m-1}} v_{r,m-1} W'_{kr,m} > 0 \right\} \prod_{r=1}^{V_{m-1}} P\left(W'_{kr,m}\right) \right] \tag{3.10}$$

and $P\left(\mathbf{v}_l|\mathbf{v}_{l-1}, W_{ij,l}\right)$, which is defined identically to $P\left(\mathbf{v}_l|\mathbf{v}_{l-1}\right)$, except that the summation is performed over all configurations in which $W_{ij,l}$ is fixed (*i.e.*, $\mathbf{W}'_l : W'_{ij,l} = W_{ij,l}$) and we set

$P(W_{ij,l}) = 1$. Now we can write recursively $P(\mathbf{v}_1) = P(\mathbf{v}_1|\mathbf{v}_0 = \mathbf{x})$

$$\forall m \in \{2, .., l-1\} : P(\mathbf{v}_m) = \sum_{\mathbf{v}_{m-1}} P(\mathbf{v}_m|\mathbf{v}_{m-1}) P(\mathbf{v}_{m-1}) \tag{3.11}$$

$$P(\mathbf{v}_l|W_{ij,l}) = \sum_{\mathbf{v}_{l-1}} P(\mathbf{v}_l|\mathbf{v}_{l-1}, W_{ij,l}) P(\mathbf{v}_{l-1}) \tag{3.12}$$

$$\forall m \in \{l+1, l+2, .., L\} : P(\mathbf{v}_m|W_{ij,l}) = \sum_{\mathbf{v}_{m-1}} P(\mathbf{v}_m|\mathbf{v}_{m-1}) P(\mathbf{v}_{m-1}|W_{ij,l}) \tag{3.13}$$

Thus we obtain the result of Eq. 3.9, through $P(\mathbf{y}|W_{ij,l}) = P(\mathbf{v}_L = \mathbf{y}|W_{ij,l})$. However, this computation is still generally intractable, since all of the above summations (Eqs. 3.10-3.13) are still over an exponential number of values. Therefore, we need to make one additional approximation.

### 3.4 Approximation 2: large fan-in

Next we simplify the above summations (Eqs. 3.10-3.13) assuming that the neuronal fan-in is "large". We keep in mind that $i, j$ and $l$ are the specific indices of the fixed weight $W_{ij,l}$. All the other weights beside $W_{ij,l}$ can be treated as independent random variables, due to the mean field approximation (Eq. 3.5). Therefore, in the limit of a infinite neuronal fan-in ($\forall m : K_m \to \infty$) we can use the Central Limit Theorem (CLT) and say that the normalized input to each neuronal layer, is distributed according to a Gaussian distribution

$$\forall m : \mathbf{u}_m = \mathbf{W}_m \mathbf{v}_{m-1}/\sqrt{K_m} \sim \mathcal{N}(\boldsymbol{\mu}_m, \boldsymbol{\Sigma}_m) . \tag{3.14}$$

Since $K_m$ is actually finite, this would be only an approximation - though a quite common and effective one (*e.g.*, [22]). Using the approximation in Eq. 3.14 together with $\mathbf{v}_m = \mathrm{sign}(\mathbf{u}_m)$ (Eq. 2.1) we can calculate (appendix B) the distribution of $\mathbf{u}_m$ and $\mathbf{v}_m$ sequentially for all the layers $m \in \{1, \dots, L\}$, for any given value of $\mathbf{v}_0$ and $W_{ij,l}$. These effectively simplify the summations in 3.10-3.13 using Gaussian integrals (appendix B).

At the end of this "forward pass" we will be able to find $P(\mathbf{y}|W_{ij,l}) = P(\mathbf{v}_L = \mathbf{y}|W_{ij,l}), \forall i, j, l$. This takes a polynomial number of steps (appendix B.3), instead of a direct calculation through Eqs. 3.11-3.13, which is exponentially hard. Using $P(\mathbf{y}|W_{ij,l})$ and Eq. 3.7 we can now update the distribution of $P(W_{ij,l})$. This immediately gives the Bayes estimate of the weights (Eq. 3.6) and outputs (Eq. 3.2).

As we note in appendix B.3, the computational complexity of the forward pass is significantly lower if $\boldsymbol{\Sigma}_m$ is diagonal. This is true exactly only in special cases. For example, this is true if all hidden neurons have a fan-out of one - such as in a 2-layer network with a single output. However, in order to reduce the computational complexity in cases that $\boldsymbol{\Sigma}_m$ is not diagonal, we will approximate the distribution of $\mathbf{u}_m$ with its factorized ('mean-field') version. Recall that the optimal factor is the marginal of the distribution (appendix A). Therefore, we can now find $P(\mathbf{y}|W_{ij,l})$ easily (appendix B.1), as all the off-diagonal components in $\boldsymbol{\Sigma}_m$ are zero, so $\boldsymbol{\Sigma}_{kk',m} = \sigma_{k,m}^2 \delta_{kk'}$.

A direct calculation of $P(\mathbf{v}_L = \mathbf{y}|W_{ij,l})$ for every $i, j, l$ would be computationally wasteful, since we will repeat similar calculations many times. In order to improve the algorithm's efficiency, we again exploit the fact that $K_l$ is large. We approximate $\ln P(\mathbf{v}_L = \mathbf{y}|W_{ij,l})$ using a Taylor expansion of $W_{ij,l}$ around its mean, $\langle W_{ij,l} \rangle$, to first order in $K_l^{-1/2}$. The first order terms in this expansion can be calculated using backward propagation of derivative terms

$$\Delta_{k,m} = \partial \ln P(\mathbf{v}_L = \mathbf{y})/\partial \mu_{k,m} , \tag{3.15}$$

similarly to the BP algorithm (appendix C). Thus, after a forward pass for $m = 1, \dots, L$, and a backward pass for $l = L, \dots, 1$, we obtain $P(\mathbf{v}_L = \mathbf{y}|W_{ij,l})$ for all $W_{ij,l}$ and update $P(W_{ij,l})$.

## 4 The Expectation Backpropagation Algorithm

Using our results we can efficiently update the posterior distribution $P(W_{ij,l}|D_n)$ for all the weights with $O(|\mathcal{W}|)$ operations, according to Eqs. 3.7. Next, we summarize the resulting general algorithm - the Expectation BackPropgation (EBP) algorithm. In appendix D, we exemplify how to apply the

algorithm in the special cases of MNNs with binary, ternary or real (continuous) weights. Similarly to the original BP algorithm (see review in [16]), given input $\mathbf{x}$ and desired output $\mathbf{y}$, first we perform a forward pass to calculate the mean output $\langle \mathbf{v}_l \rangle$ for each layer. Then we perform a backward pass to update $P\left(W_{ij,l}|D_n\right)$ for all the weights.

**Forward pass** In this pass we perform the forward calculation of probabilities, as in Eq. 3.11. Recall that $\langle W_{kr,m} \rangle$ is the mean of the posterior distribution $P\left(W_{kr,m}|D_n\right)$. We first initialize the MNN input $\langle v_{k,0} \rangle = x_k$ for all $k$ and calculate recursively the following quantities for $m = 1, \ldots, L$ and all $k$

$$\mu_{k,m} = \frac{1}{\sqrt{K_m}} \sum_{r=1}^{V_{m-1}} \langle W_{kr,m} \rangle \langle v_{r,m-1} \rangle \quad ; \quad \langle v_{k,m} \rangle = 2\Phi\left(\mu_{k,m}/\sigma_{k,m}\right) - 1 \,. \tag{4.1}$$

$$\sigma_{k,m}^2 = \frac{1}{K_m} \sum_{r=1}^{V_{m-1}} \langle W_{kr,m}^2 \rangle \left(\delta_{m,1}\left(\langle v_{r,m-1} \rangle^2 - 1\right) + 1\right) - \langle W_{kr,m} \rangle^2 \langle v_{r,m-1} \rangle^2 \,, \tag{4.2}$$

where $\boldsymbol{\mu}_m$ and $\boldsymbol{\sigma}_m^2$ are, respectively, the mean and variance of $\mathbf{u}_m$, the input of layer $m$ (Eq. 3.14), and $\langle \mathbf{v}_m \rangle$ is the resulting mean of the output of layer $m$.

**Backward pass** In this pass we perform the Bayes update of the posterior (Eq. 3.7) using a Taylor expansion. Recall Eq. 3.15. We first initialize[4]

$$\Delta_{i,L} = y_i \frac{\mathcal{N}\left(0|\mu_{i,L}, \sigma_{i,L}^2\right)}{\Phi\left(y_i \mu_{i,L}/\sigma_{i,L}\right)} \,. \tag{4.3}$$

for all $i$. Then, for $l = L, \ldots, 1$ and $\forall i, j$ we calculate

$$\Delta_{i,l-1} = \frac{2}{\sqrt{K_l}} \mathcal{N}\left(0|\mu_{i,l-1}, \sigma_{i,l-1}^2\right) \sum_{j=1}^{V_m} \langle W_{ji,l} \rangle \Delta_{j,l} \,. \tag{4.4}$$

$$\ln P\left(W_{ij,l}|D_n\right) = \ln P\left(W_{ij,l}|D_{n-1}\right) + \frac{1}{\sqrt{K_l}} W_{ij,l} \Delta_{i,l} \langle v_{j,l-1} \rangle + C \,, \tag{4.5}$$

where $C$ is some unimportant constant (which does not depend on $W_{ij,l}$).

**Output** Using the posterior distribution, the optimal configuration can be immediately found through the MAP weights estimate (Eq. 3.6) $\forall i, j, l$

$$W_{ij,l}^* = \operatorname{argmax}_{W_{ij,l} \in S_{ij,l}} \ln P\left(W_{ij,l}|D_n\right) \,. \tag{4.6}$$

The output of a MNN implementing these weights would be $g\left(\mathbf{x}, \mathcal{W}^*\right)$ (see Eq. 2.2). We define this to be the 'deterministic' EBP output (EBP-D).

Additionally, the MAP output (Eq. 3.2) can be calculated directly

$$\mathbf{y}^* = \operatorname{argmax}_{\mathbf{y} \in \mathcal{Y}} \ln P\left(\mathbf{v}_L = \mathbf{y}\right) = \operatorname{argmax}_{\mathbf{y} \in \mathcal{Y}} \left[\sum_k \ln \left(\frac{1 + \langle v_{k,L} \rangle}{1 - \langle v_{k,L} \rangle}\right)^{y_k}\right] \tag{4.7}$$

using $\langle v_{k,L} \rangle$ from Eq. 4.1, or as an ensemble average over the outputs of all possible MNN with the weights $W_{ij,l}$ being sampled from the estimated posterior $P\left(W_{ij,l}|D_n\right)$. We define the output in Eq. 4.7 to be the Probabilistic EBP output (EBP-P). Note that in the case of a single output $\mathcal{Y} = \{-1, 1\}$, so this output simplifies to $y = \operatorname{sign}\left(\langle v_{k,L} \rangle\right)$.

$$\Delta_{i,L} = -\frac{\mu_{i,L}}{\sigma_{i,L}^2 \sqrt{K_L}} \mathcal{I}\left\{y_i \mu_{i,L} < 0\right\}$$

# 5 Numerical Experiments

We use several high dimensional text datasets to assess the performance of the EBP algorithm in a supervised binary classification task. The datasets (taken from [7]) contain eight binary tasks from four datasets: 'Amazon (sentiment)', '20 Newsgroups', 'Reuters' and 'Spam or Ham'. Data specification ($N$ =#examples and $M$ =#features) and results (for each algorithm) are described in Table 1. More details on the data including data extraction and labeling can be found in [7].

We test the performance of EBP on MNNs with a 2-layer architecture of $M \rightarrow 120 \rightarrow 1$, and bias weights. We examine two special cases: (1) MNNs with real weights (2) MNNs with binary weights (and real bias). Recall the motivation for the latter (section 1) is that they can be efficiently implemented in hardware (real bias has negligible costs). Recall also that for each type of MNN, the algorithm gives two outputs - EBP-D (deterministic) and EBP-P (probabilistic), as explained near Eqs. 4.6-4.7.

To evaluate our results we compare EBP to: (1) the AROW algorithm, which reports state-of-the-art results on the tested datasets [7] (2) the traditional Backpropagation (BP) algorithm, used to train an $M \rightarrow 120 \rightarrow 1$ MNN with real weights. In the latter case, we used both Cross Entropy (CE) and Mean Square Error (MSE) as loss functions. On each dataset we report the results of BP with the loss function which achieved the minimal error. We use a simple parameter scan for both AROW (regularization parameter) and the traditional BP (learning rate parameter). Only the results with the optimal parameters (*i.e.*, achieving best results) are reported in Table 1. The optimal parameters found were never at the edges of the scanned field. Lastly, to demonstrate the destructive effect of naive quantization, we also report the performance of the BP-trained MNNs, after all the weights (except the bias) were clipped using a sign function.

During training the datasets were repeatedly presented in three epochs (in all algorithms, additional epochs did not reduce test error). On each epoch the examples were shuffled at random order for BP and EBP (AROW determines its own order). The test results are calculated after each epoch using 8-fold cross-validation, similarly to [7]. Empirically, EBP running time is similar to BP with real weights, and twice slower with binary weights. For additional implementation details, see appendix E.1. The code is available on the author's website.

The minimal values achieved over all three epochs are summarized in Table 1. As can be seen, in all datasets EBP-P performs better then AROW, which performs better then BP. Also, EBP-P usually perfroms better with binary weights. In appendix E.2 we show that this ranking remains true even if the fan-in is small (in contrast to our assumptions), or if a deeper 3-layer architecture is used.

| Dataset | #Examples | #Features | Real EBP-D | Real EBP-P | Binary EBP-D | Binary EBP-P | AROW | BP | Clipped BP |
|---|---|---|---|---|---|---|---|---|---|
| Reuters news I6 | 2000 | 11463 | 14.5% | 11.35% | 21.7% | **9.95%** | 11.72% | 13.3% | 26.15% |
| Reuters news I8 | 2000 | 12167 | 15.65% | **15.25%** | 23.15% | 16.4% | 15.27% | 18.2% | 26.4% |
| Spam or ham d0 | 2500 | 26580 | 1.28% | 1.11% | 7.93% | **0.76%** | 1.12% | 1.32% | 7.97% |
| Spam or ham d1 | 2500 | 27523 | 1.0% | **0.96%** | 3.85% | **0.96%** | 1.4% | 1.36% | 7.33% |
| 20News group comp vs HW | 1943 | 29409 | 5.06% | 4.96% | 7.54% | **4.44%** | 5.79% | 7.02% | 13.07% |
| 20News group elec vs med | 1971 | 38699 | 3.36% | 3.15% | 6.0% | **2.08%** | 2.74% | 3.96% | 14.23% |
| Amazon Book reviews | 3880 | 221972 | 2.14% | 2.09% | 2.45% | **2.01%** | 2.24% | 2.96% | 3.81% |
| Amazon DVD reviews | 3880 | 238739 | **2.06%** | 2.14% | 5.72% | 2.27% | 2.63% | 2.94% | 5.15% |

Table 1: Data specification, and test errors (with 8-fold cross-validation). Best results are boldfaced.

# 6 Discussion

Motivated by the recent success of MNNs, we developed the Expectation BackPropagation algorithm (EBP - see section 4) for approximate Bayesian inference of the synaptic weights of a MNN. Given a supervised classification task with labeled training data and a prior over the weights, this deterministic online algorithm can be used to train deterministic MNNs (Eq. 2.2) without the need to tune learning parameters (*e.g.*, learning rate). Furthermore, each synaptic weight can be restricted to some set - which can be either finite (*e.g.*, binary numbers) or infinite (*e.g.*, real numbers). This opens the possibility of implementing trained MNNs in power-efficient hardware devices requiring limited parameter precision.

This algorithm is essentially an analytic approximation to the intractable Bayes calculation of the posterior distribution of the weights after the arrival of a new data point. To simplify the intractable Bayes update rule we use several approximations. First, we approximate the posterior using a product of its marginals - a 'mean field' approximation. Second, we assume the neuronal layers have a large fan-in, so we can approximate them as Gaussian. After these two approximations each Bayes update can be tractably calculated in polynomial time in the size of the MNN. However, in order to further improve computational complexity (to $O(|\mathcal{W}|)$ in each step, like BP), we make two additional approximations. First, we use the large fan-in to perform a first order expansion. Second, we optionally[5] perform a second 'mean field' approximation - to the distribution of the neuronal inputs. Finally, after we obtain the approximated posterior using the algorithm, the Bayes estimates of the most probable weights and the outputs are found analytically.

Previous approaches to obtain these Bayes estimates were too limited for our purposes. The Monte Carlo approach [21] achieves state-of-the-art performance for small MNNs [26], but does not scale well [25]. The Laplace approximation [17] and variational Bayes [10, 2, 9] based methods require real-value weights, tuning of the learning rate parameter, and stochastic neurons (to "smooth" the likelihood). Previous EP [24, 22] and message passing [6, 1] (a special case of EP[5]) based methods were derived only for SNNs.

In contrast, the EBP allows parameter free and scalable training of various types of MNNs (deterministic or stochastic) with discrete (*e.g.*, binary) or continuous weights. In appendix F, we see that for continuous weights EBP is almost identical to standard BP with a specific choice of activation function $s(x) = 2\Phi(x) - 1$, CE loss and learning rate $\eta = 1$. The only difference is that the input is normalized by its standard deviation (Eq. 4.1, *right*), which depends on the weights and inputs (Eq. 4.2). This re-scaling makes the learning algorithm invariant to the amplitude changes in the neuronal input. This results from the same invariance of the sign activation functions. Note that in standard BP algorithm the performance is directly affected by the amplitude of the input, so it is a recommended practice to re-scale it in pre-processing [16].

We numerically evaluated the algorithm on binary classification tasks using MNNs with two or three synaptic layers. In all data sets and MNNs EBP performs better than standard BP with the optimal constant learning rate, and even achieves state-of-the-art results in comparison to [7]. Surprisingly, EBP usually performs best when it is used to train binary MNNs. As suggested by a reviewer, this could be related to the type of problems examined here. In text classification tasks have large sparse input spaces (bag of words), and presence/absence of features (words) is more important than their real values (frequencies). Therefore, (distributions over) binary weights and a threshold activation function may work well.

In order to get such a good performance in binary MNNs, one must average over the output the inferred (approximate) posterior of the weights. The EBP-P output of the algorithm calculates this average analytically. In hardware this output could be realizable by averaging the output of several binary MNNs, by sampling weights from $P(W_{ij,l}|D_n)$. This can be done efficiently (appendix G).

Our numerical testing mainly focused on high-dimensional text classification tasks, where shallow architectures seem to work quite well. In other domains, such as vision [14] and speech [8], deep architectures achieve state-of-the-art performance. Such deep MNNs usually require considerable fine-tuning and additional 'tricks' such as unsupervised pre-training [8], weight sharing [14] or momentum[6]. Integrating such methods into EBP and using it to train deep MNNs is a promising direction for future work. Another important generalization of the algorithm, which is rather straightforward, is to use activation functions other than $\text{sign}(\cdot)$. This is particularly important for the last layer - where a linear activation function would be useful for regression tasks, and joint activation functions[7] would be useful for multi-class tasks[4].

**Acknowledgments** The authors are grateful to C. Baldassi, A. Braunstein and R. Zecchina for helpful discussions and to A. Hallak, T. Knafo and U. Sümbül for reviewing parts of this manuscript. The research was partially funded by the Technion V.P.R. fund, by the Intel Collaborative Research Institute for Computational Intelligence (ICRI-CI), and by the Gruss Lipper Charitable Foundation.

## Footnotes

[1]*i.e.*, having more than a single layer of adjustable weights.

[2]*i.e.*, having only a single layer of adjustable weights.

[3]MNN with stochastic activation functions will have a "smoothed out" version of this.

[4]Due to numerical inaccuracy, calculating $\Delta_{i,L}$ using Eq. 4.3 can generate nonsensical values ($\pm\infty$, NaN) if $|\mu_{i,L}/\sigma_{i,L}|$ becomes to large. If this happens, we use instead the asymptotic form in that limit

[5]This approximation is not required if all neurons in the MNN have a fan-out of one.

[6]Which departs from the online framework considered here, since it requires two samples in each update.

[7]*i.e.*, activation functions for which $(f(\mathbf{x}))_i \neq f(x_i)$, such as softmax or argmax.

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
