[Supplementary Material]

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

1. We use the Bayes update (Eq. 3.3) with $\hat{P}\left(\mathcal{W}|D_{n-1}\right)$ as our prior

$$
\begin{aligned}
\tilde{P}\left(\mathcal{W}|D_n\right) &\propto& P\left(\mathbf{y}^{(n)}|\mathbf{x}^{(n)}, \mathcal{W}\right) \hat{P}\left(\mathcal{W}|D_{n-1}\right) \\
&=& P\left(\mathbf{y}^{(n)}|\mathbf{x}^{(n)}, \mathcal{W}\right) \prod_{i,j,l} \hat{P}\left(W_{ij,l}|D_{n-1}\right) ,
\end{aligned}
\tag{A.1}
$$

   where $\tilde{P}\left(\mathcal{W}|D_n\right)$ is some "temporary" posterior distribution.

2. We project $\hat{P}\left(\mathcal{W}|D_n\right)$ onto $\tilde{P}\left(\mathcal{W}|D_n\right)$ by minimizing the reverse Kullback-Leibler divergence (e.g., as in the expectation propagation algorithm [25, 7])

$$D_{KL}\left(\tilde{P}\left(\mathcal{W}|D_n\right) || \hat{P}\left(\mathcal{W}|D_n\right)\right) = \sum_{\mathcal{W}} \tilde{P}\left(\mathcal{W}|D_n\right) \log\left(\frac{\tilde{P}\left(\mathcal{W}|D_n\right)}{\hat{P}\left(\mathcal{W}|D_n\right)}\right)$$

   with the normalization constraint $\sum_{W_{ij,l}} \hat{P}\left(W_{ij,l}|D_n\right) = 1 \; \forall i, j, l$.

The second step can be easily performed using Lagrange multipliers, forming a Lagrangian

$$
\begin{aligned}
L\left(\hat{P}\left(\mathcal{W}|D_n\right)\right) &=& \sum_{\mathcal{W}' \in \mathcal{S}} \tilde{P}\left(\mathcal{W}'|D_n\right) \log\left(\frac{\tilde{P}\left(\mathcal{W}'|D_n\right)}{\prod_{k,r,m} \hat{P}\left(W'_{kr,m}|D_n\right)}\right) \\
&& + \sum_{k,r,m} \lambda_{kr,m} \left(1 - \sum_{W'_{kr,m} \in S_{kr,m}} \hat{P}\left(W'_{kr,m}|D_n\right)\right) .
\end{aligned}
$$

The minimum is found by differentiating and equating to zero

$$0 = \frac{\partial L\left(\hat{P}\left(\mathcal{W}|D_n\right)\right)}{\partial \hat{P}\left(W_{ij,l}|D_n\right)} = -\frac{\sum_{\mathcal{W}':W'_{ij,l}=W_{ij,l}} \tilde{P}\left(\mathcal{W}'|D_n\right)}{\hat{P}\left(W_{ij,l}|D_n\right)} - \lambda_{ij,l} .$$

Using this equation together with the normalization constraint $\sum_{W_{ij,l}} \hat{P}\left(W_{ij,l}|D_n\right) = 1 \; \forall i, j, l$ we obtain the result of the minimization through marginalization

$$\hat{P}\left(W_{ij,l}|D_n\right) = \sum_{\mathcal{W}':W'_{ij,l}=W_{ij,l}} \tilde{P}\left(\mathcal{W}'|D_n\right) , \tag{A.2}$$

which is a known result [7, p. 468]. Finally, we can combine step 1 (Bayes update) with step 2 (projection) to a single step

$$\hat{P}\left(W_{ij,l}|D_n\right) = \sum_{\mathcal{W}':W'_{ij,l}=W_{ij,l}} P\left(\mathbf{y}^{(n)}|\mathbf{x}^{(n)}, \mathcal{W}'\right) \prod_{k,r,m} \hat{P}\left(W'_{kr,m}|D_{n-1}\right) .$$

This step is exactly Eqs. 3.7 and 3.8 combined.

# B  Forward propagation of probabilities

In this section we simplify the summations in Eqs. (3.10-3.13), by assuming that the fan-in of all of the connections is "large", *i.e.*, $\forall m : K_m \to \infty$. Using this approximation and the CLT we can write $\forall m \geq 1$

$$\mathbf{u}_m = \frac{1}{\sqrt{K_m}} \mathbf{W}_m \mathbf{v}_{m-1} \sim \mathcal{N}\left(\boldsymbol{\mu}_m, \boldsymbol{\Sigma}_m\right) . \tag{B.1}$$

$$\mathbf{v}_m = \text{sign}\left(\mathbf{u}_m\right) \tag{B.2}$$

with $\mathbf{v}_0 = \mathbf{x}$. Recall (from Eq. 3.9) that $W_{ij,l}$ is a specific weight which is fixed (so $i, j$ and $l$ are "special" indexes), while all the other weights $W_{kr,m}$ (for which $k \neq i$ or $r \neq j$ or $m \neq l$) are independent variables. We first consider a simple special case.

## B.1  Special case: a diagonal $\boldsymbol{\Sigma}_m$

In this section we assume initially that $\boldsymbol{\Sigma}_m$ in Eq. B.1 is diagonal, so

$$P\left(\mathbf{u}_m\right) = \prod_k \mathcal{N}\left(u_{k,m}|\mu_{k,m}, \sigma_{k,m}^2\right) \tag{B.3}$$

with $\sigma_{k,m}^2 = \Sigma_{kk,m}$. Therefore, $\forall m \in \{1, \ldots, l-1\}$, we can use Eq. B.2 to obtain

$$P\left(\mathbf{v}_m\right) = \prod_k P\left(v_{k,m}\right) = \prod_k \Phi\left(v_{k,m}\mu_{k,m}/\sigma_{k,m}\right) . \tag{B.4}$$

These distributions, which are the approximate solution of Eqs. 3.11-3.13, immediately give

$$\langle v_{k,m}\rangle = v_{k,0}\delta_{0m} + (1-\delta_{0m})\left(2\Phi\left(\mu_{k,m}/\sigma_{k,m}\right) - 1\right) . \tag{B.5}$$

Note that $W_{kr,m}$ and $v_{k,m-1}$ are independent for a fixed $m$ (from Eqs. 3.5 and 2.1). Therefore, it is straightforward to derive

$$\mu_{k,m} = \langle u_{k,m}\rangle = \frac{1}{\sqrt{K_m}} \sum_{r=1}^{V_{m-1}} \langle W_{kr,m}\rangle \langle v_{r,m-1}\rangle \tag{B.6}$$

$$\sigma_{k,m}^2 = \text{Var}\left(u_{k,m}\right) = \frac{1}{K_m} \sum_{r=1}^{V_{m-1}} \langle W_{kr,m}^2\rangle \langle v_{r,m-1}^2\rangle - \langle W_{kr,m}\rangle^2 \langle v_{r,m-1}\rangle^2 . \tag{B.7}$$

Since the value $\mathbf{v}_0$ is given ($\mathbf{v}_0 = \mathbf{x}$), and for $m \geq 1$, $\mathbf{v}_m$ are binary vectors, we have $\langle v_{r,m-1}^2\rangle = 1 + \delta_{m0}\left(v_{r,0}^2 - 1\right)$. Importantly, if we know $\mathbf{x}$ and $P\left(W_{kr,m}\right)$ Eqs. B.5-B.7 can be calculated together in a sequential "forward pass" for $m = 1, 2, ..., l-1$. We continue in the same manner for $m \geq l$, with slight modifications, since $W_{ij,l}$ is fixed. For $m = l$ we need to respectively replace $\langle W_{ij,l}\rangle$ and $\langle W_{ij,l}^2\rangle$ with fixed values $W_{ij,l}$ and $W_{ij,l}^2$ so

$$\mu_{i,l}\left(W_{ij,l}\right) = \mu_{i,l} + \frac{1}{\sqrt{K_l}}\left(W_{ij,l} - \langle W_{ij,l}\rangle\right)\langle v_{j,l-1}\rangle \tag{B.8}$$

$$\sigma_{i,l}^2\left(W_{ij,l}\right) = \sigma_{i,l}^2 - \frac{1}{K_l}\langle W_{ij,l}\rangle^2 \text{Var}\left(v_{j,l-1}\right) . \tag{B.9}$$

For $m > l$ we continue as in Eqs. B.4-B.7, except we replace $P\left(v_{k,m}\right), \langle v_{k,m}\rangle, \mu_{k,m}$ and $\sigma_{k,m}^2$ with $P\left(v_{k,m}|W_{ij,l}\right), \langle v_{k,m}|W_{ij,l}\rangle, \mu_{k,m}\left(W_{ij,l}\right)$ and $\sigma_{k,m}^2\left(W_{ij,l}\right)$, respectively, to emphasize that they depend on $W_{ij,l}$. The end result of this calculation is $P\left(\mathbf{v}_L = \mathbf{y}|W_{ij,l}\right)$.

## B.2  General case: non-diagonal $\boldsymbol{\Sigma}_m$

In this section we perform the forward propagation (Eqs. 3.11-3.13) without assuming that $\boldsymbol{\Sigma}_m$ is diagonal. For simplicity, in this section we will usually suppress explicit dependence on $W_{ij,l}$ in our notation (keeping in mind that we should respectively replace $\langle W_{ij,l}\rangle$ and $\langle W_{ij,l}^2\rangle$ with $W_{ij,l}$ and $W_{ij,l}^2$).

Using these equations it is straightforward to derive $\boldsymbol{\mu}_m$ and $\boldsymbol{\Sigma}_m$ for each layer and $\forall k, k'$

$$\mu_{k,m} = \frac{1}{\sqrt{K_m}} \sum_{r=1}^{V_{m-1}} \langle W_{kr,m} \rangle \langle v_{r,m-1} \rangle \tag{B.10}$$

$$\Sigma_{kk',m} = \text{Cov}\left(u_{k,m}, u_{k',m}\right) \tag{B.11}$$

$$= \frac{1}{K_m} \delta_{kk'} \sum_{r=1}^{V_{m-1}} \text{Var}\left(W_{kr,m}\right) \langle v_{r,m-1}^2 \rangle$$

$$+ \frac{1}{K_m} \sum_{r=1}^{V_{m-1}} \sum_{r'=1}^{V_{m-1}} \langle W_{kr,m} \rangle \langle W_{k'r',m} \rangle \text{Cov}\left(v_{r,m-1}, v_{r',m-1}\right).$$

Therefore, for $m > 1$ and $\forall k$

$$\begin{aligned}
\langle v_{k,m} \rangle &= P\left(u_{k,m} > 0\right) - P\left(u_{k,m} < 0\right) \\
&= 2P\left(u_{k,m} > 0\right) - 1 \\
&= 2\Phi\left(\mu_{k,m}/\Sigma_{kk,m}\right) - 1
\end{aligned} \tag{B.12}$$

and $\forall k' \neq k$ :

$$\begin{aligned}
\langle v_{k,m} v_{k',m} \rangle &= P\left(\text{sign}\left(u_{k,m} u_{k',m}\right) > 0\right) - P\left(\text{sign}\left(u_{k,m} u_{k',m}\right) < 0\right) \\
&= P\left(u_{k,m} > 0, u_{k',m} > 0\right) + P\left(u_{k,m} < 0, u_{k',m} < 0\right) \\
&\quad - P\left(u_{k,m} > 0, u_{k',m} < 0\right) - P\left(u_{k,m} < 0, u_{k',m} > 0\right) \\
&= 2P\left(u_{k,m} > 0, u_{k',m} > 0\right) + 2P\left(u_{k,m} < 0, u_{k',m} < 0\right) - 1.
\end{aligned}$$

Note that for $m \geq l$, all these results depend on $W_{ij,l}$ (this dependency was suppressed, for brevity). Lastly, we obtain

$$P\left(\mathbf{y}|W_{ij,l}\right) = P\left(\forall k : u_{k,L} y_k > 0\right). \tag{B.13}$$

All that remains is to find is to substitute $P\left(\mathbf{y}|W_{ij,l}\right)$ into Eq. 3.7 and perform the update rule.

### B.3 Computational Complexity

What is the computational complexity of a direct implementation the resulting update rule for each weight? We first consider the complexity for the general case of a non-Diagonal $\boldsymbol{\Sigma}_m$. Denoting $S = \max_{i,j,l} |S_{ij,l}|$, $V = \max_l V_l$ and $\epsilon$ as the required (relative) precision for calculating $P\left(\mathbf{y}|W_{ij,l}\right)$, we can now find the worst case asymptotic complexity $\tilde{O}\left(\cdot\right)$ (*i.e.*, neglecting logarithmic factors) of all steps in the update rule. To do this, we first note that

- Given $P\left(W_{ij,l}\right)$, calculating $\langle W_{ij,l} \rangle$ is $O\left(S\right)$.
- Calculating $\Phi\left(x\right)$ is $\tilde{O}\left(1\right)$ [9].
- The current state-of-the-art complexity of calculating Gaussian orthant probabilities in $d$ dimensions and relative precision $\epsilon$:
  - For $d \leq 3$ it is $\tilde{O}\left(1\right)$ [13].
  - For $d \geq 4$ it is $\tilde{O}\left(d^3 \epsilon^{-2}\right)$ [10], and sometimes lower (*e.g.*, see $d = 4$ in [15]).

Therefore , in the non-diagonal case (appendix B.2)

- Eq. B.10 for all layers is $O\left(S\,|\mathcal{W}|\right)$.
- Eq. B.11 for all layers is $O\left(S\,|\mathcal{W}|\,V\right)$.
- Eq. B.12 for all neurons and all layers is $\tilde{O}\left(LV\right)$.
- Eq. **??** for all neurons and all layers is $\tilde{O}\left(|\mathcal{W}|\right)$.
- Eq. B.13 is $\tilde{O}\left(V_L^3 \epsilon^{-2}\right)$ if $V_L > 3$ or $\tilde{O}\left(1\right)$ otherwise.

Summing all contributions, in the worst case, the total computational complexity of a single update step (Eq. 3.7) for all weights and weight values is $\tilde{O}\left(S^2 \left|\mathcal{W}\right|^2 V\right)$ if $V_L \leq 3$ or $\tilde{O}\left(S^2 \left|\mathcal{W}\right|^2 V + V_L^3 \epsilon^{-2} \left|\mathcal{W}\right| S\right)$ if $V_L > 3$.

If instead, $\boldsymbol{\Sigma}_m$ is diagonal, then, using a similar analysis, it is straightforward to show that computational complexity of a "naive" single update step (Eq. 3.7) for all weights and weight values is instead $O\left(S^2 \left|\mathcal{W}\right|^2\right)$. However, this complexity can be further reduced to a linear complexity $O\left(S^2 \left|\mathcal{W}\right|\right)$ by exploiting the fact that many similar operations are shared by the updates of different weights. We explain how this is done in Appendix C.

## C    Backward propagation of derivatives

In this section we calculate the Taylor expansion of $\ln P\left(\mathbf{v}_L = \mathbf{y}|W_{ij,l}\right)$ in $W_{ij,l}$ around $\langle W_{ij,l}\rangle$. Initially, we perform a similar Taylor expansion without the log. This yields, to first order

$$P\left(\mathbf{v}_L = \mathbf{y}|W_{ij,l}\right) = P\left(\mathbf{v}_L = \mathbf{y}|W_{ij,l} = \langle W_{ij,l}\rangle\right) + \left(W_{ij,l} - \langle W_{ij,l}\rangle\right)\left[\frac{\partial P\left(\mathbf{v}_L = \mathbf{y}|W_{ij,l}\right)}{\partial W_{ij,l}}\right]_{W_{ij,l}=\langle W_{ij,l}\rangle} \quad \text{(C.1)}$$

We re-write this expression to first order using the notation as in appendix B.1. First, using the chain rule, B.8 and Eq. B.9 we obtain

$$\frac{\partial P\left(\mathbf{v}_L = \mathbf{y}|W_{ij,l}\right)}{\partial W_{ij,l}} = \frac{\partial \mu_{i,l}}{\partial W_{ij,l}}\frac{\partial P\left(\mathbf{v}_L = \mathbf{y}|W_{ij,l}\right)}{\partial \mu_{i,l}} + \frac{\partial \sigma_{i,l}^2}{\partial W_{ij,l}}\frac{\partial P\left(\mathbf{v}_L = \mathbf{y}|W_{ij,l}\right)}{\partial \sigma_{i,l}^2}$$

$$= \frac{1}{\sqrt{K_l}}\langle v_{j,l-1}\rangle \frac{\partial}{\partial \mu_{i,l}}P\left(\mathbf{v}_L = \mathbf{y}|W_{ij,l}\right) . \quad \text{(C.2)}$$

Next, from Eqs. B.8-B.9

$$\mu_{i,l}\left(W_{ij,l} = \langle W_{ij,l}\rangle\right) = \mu_{i,l} \quad \text{(C.3)}$$
$$\sigma_{i,l}^2\left(W_{ij,l}\right) = \sigma_{i,l}^2 + O\left(K_l^{-1}\right) \quad \text{(C.4)}$$

and therefore
$$P\left(\mathbf{v}_L = \mathbf{y}|W_{ij,l} = \langle W_{ij,l}\rangle\right) = P\left(\mathbf{v}_L = \mathbf{y}\right) + O\left(K_l^{-1}\right) \quad \text{(C.5)}$$

Putting Eqs. C.1-C.5 together, we obtain

$$P\left(\mathbf{v}_L = \mathbf{y}|W_{ij,l}\right) = P\left(\mathbf{v}_L = \mathbf{y}\right) + \frac{1}{\sqrt{K_l}}\left(W_{ij,l} - \langle W_{ij,l}\rangle\right)\frac{\partial P\left(\mathbf{v}_L = \mathbf{y}\right)}{\partial \mu_{i,l}}\langle v_{j,l-1}\rangle + O\left(K_l^{-1}\right) .$$

Taking the logarithm of this expression, we obtain to first order

$$\ln P\left(\mathbf{v}_L = \mathbf{y}|W_{ij,l}\right) = \ln\left[P\left(\mathbf{v}_L = \mathbf{y}\right) + \frac{1}{\sqrt{K_l}}\left(W_{ij,l} - \langle W_{ij,l}\rangle\right)\frac{\partial P\left(\mathbf{v}_L = \mathbf{y}\right)}{\partial \mu_{i,l}}\langle v_{j,l-1}\rangle + O\left(K_l^{-1}\right)\right]$$

$$= \ln\left[1 + \frac{1}{\sqrt{K_l}}\langle v_{j,l-1}\rangle\left(W_{ij,l} - \langle W_{ij,l}\rangle\right)\frac{\partial P\left(\mathbf{v}_L = \mathbf{y}\right)}{\partial \mu_{i,l}}/P\left(\mathbf{v}_L = \mathbf{y}\right) + O\left(K_l^{-1}\right)\right] + C$$

$$= C + \frac{1}{\sqrt{K_l}}W_{ij,l}\frac{\partial \ln P\left(\mathbf{v}_L = \mathbf{y}\right)}{\partial \mu_{i,l}}\langle v_{j,l-1}\rangle + O\left(K_l^{-1}\right) . \quad \text{(C.6)}$$

where $C$ is some constant that does not depend on $W_{ij,l}$.

As we show next, the derivative term can be calculated efficiently to first order, using the chain rule. From Eq. B.4, we have

$$\frac{\partial \ln P\left(\mathbf{v}_L = \mathbf{y}\right)}{\partial \mu_{k,L}} = \frac{\partial}{\partial \mu_{k,L}}\left[\sum_{r=1}^{V_L} \ln \Phi\left(y_r \mu_{r,L}/\sigma_{r,L}\right)\right] \quad \text{(C.7)}$$

$$= y_k \frac{\mathcal{N}\left(0|\mu_{k,L}, \sigma_{k,L}^2\right)}{\Phi\left(y_k \mu_{k,L}/\sigma_{k,L}\right)} , \quad \text{(C.8)}$$

where we used the fact that for $a = \pm 1$

$$\frac{d}{dx} \Phi\left(ax/b\right) = a\mathcal{N}\left(0|x, b^2\right) .$$

From Eq. B.6-B.7, we have

$$
\begin{aligned}
\frac{\partial \ln P\left(\mathbf{v}_L = \mathbf{y}\right)}{\partial \langle v_{k,m-1}\rangle} &= \sum_{r=1}^{V_m} \left[\frac{\partial \mu_{r,m}}{\partial \langle v_{k,m-1}\rangle} \frac{\partial \ln P\left(\mathbf{v}_L = \mathbf{y}\right)}{\partial \mu_{r,m}} + \frac{\partial \sigma_{r,m}^2}{\partial \langle v_{k,m-1}\rangle} \frac{\partial \ln P\left(\mathbf{v}_L = \mathbf{y}\right)}{\partial \sigma_{r,m}^2}\right] \\
&= \sum_{r=1}^{V_m} \left[\frac{1}{\sqrt{K_m}} \langle W_{rk,l}\rangle \frac{\partial \ln P\left(\mathbf{v}_L = \mathbf{y}\right)}{\partial \mu_{r,m}} + O\left(K_m^{-1}\right)\right] .
\end{aligned}
\tag{C.9}
$$

Finally, from Eq. B.5, we have

$$
\begin{aligned}
\frac{\partial \ln P\left(\mathbf{v}_L = \mathbf{y}\right)}{\partial \mu_{k,m}} &= \frac{\partial \langle v_{k,m}\rangle}{\partial \mu_{k,m}} \frac{\partial \ln P\left(\mathbf{v}_L = \mathbf{y}\right)}{\partial \langle v_{k,m}\rangle} \\
&= 2\mathcal{N}\left(0|\mu_{k,m}, \sigma_{k,m}^2\right) \frac{\partial \ln P\left(\mathbf{v}_L = \mathbf{y}\right)}{\partial \langle v_{k,m}\rangle} .
\end{aligned}
\tag{C.10}
$$

Importantly, we can calculate Eqs. C.8-C.10 and C.6 in a single backward pass (for $l = L, \ldots, 1$), after a single forward propagation of probabilities (Eqs. B.5-B.7, for $l = 1, \ldots, L$), similarly to the BP equations (Eqs. 1.4-1.6 in [22, Eqs. 1.4-1.6]). This reduces the computational complexity of the algorithm to $O\left(S|\mathcal{W}|\right)$.

## D    Examples for weight restrictions

In this section we explain how to implement the EBP algorithm for: (1) Binary weights (2) Ternary weights (3) Real-valued weights. Note that, similarly to BP, the algorithm uses $O\left(|\mathcal{W}|\right)$ computation steps for each update of the posterior (which is the minimal amount of steps required for any algorithm that updates all the weights) in all these examples. This computational complexity is retained as long as the restriction sets $(S_{ij,l})$ are finite, and even in some cases when they are not finite (e.g., section D.3). For simplicity, we denote in this section $\nu_{k,l} = \langle v_{k,l}\rangle$.

### D.1    Binary weights

Suppose $W_{ij,l}$ can assume only binary $\pm 1$ values, so $S_{ij,l} = \{-1, 1\}$. For convenience, we will parametrize the distribution of $W_{ij,l}$ so that

$$P\left(W_{ij,l}|D_n\right) = \frac{e^{h_{ij,l}^{(n)} W_{ij,l}}}{e^{h_{ij,l}^{(n)}} + e^{-h_{ij,l}^{(n)}}} .
\tag{D.1}$$

In the forward pass (Eq. 4.1-4.2), we can use this parametrization to compute $\langle W_{ij,l}\rangle = \tanh\left(h_{ij,l}\right)$, $\langle W_{ij,l}^2\rangle = 1$ and $\mathrm{Var}\left(W_{ij,l}\right) = 1 - \tanh^2\left(h_{ij,l}\right) = \mathrm{sech}^2\left(h_{ij,l}\right)$. In the backward pass we substitute Eq. D.1 into Eq. 4.5, and find that the parameter $h_{ij,l}^{(n)}$ should be incremented each time according to

$$h_{ij,l}^{(n)} = h_{ij,l}^{(n-1)} + \frac{1}{\sqrt{K_l}} \Delta_{i,l} \nu_{j,l-1} .
\tag{D.2}$$

Finally, we note the MAP estimate (Eq. 4.6) of the weight configuration for the MNN is obtained by simple clipping

$$W_{ij,l}^* = \mathrm{sign}\left(h_{ij,l}\right) .
\tag{D.3}$$

### D.2    Ternary weights

Suppose $S_{ij,l} = \{-1, 0, 1\}$. For convenience, we will parametrize the distribution of $W_{ij,l}$ so that

$$P\left(W_{ij,l}|D_n\right) = \frac{\exp\left(W_{ij,l} h_{ij,l}^{(n)} + \left(W_{ij,l}^2 - 1\right) g_{ij,l}^{(n)}\right)}{e^{h_{ij,l}^{(n)}} + e^{-h_{ij,l}^{(n)}} + e^{-g_{ij,l}^{(n)}}}
\tag{D.4}$$

In the forward pass

$$\langle W_{ij,l} \rangle = \frac{e^{h_{ij,l}^{(n)}} - e^{-h_{ij,l}^{(n)}}}{e^{h_{ij,l}^{(n)}} + e^{-h_{ij,l}^{(n)}} + e^{-g_{ij,l}^{(n)}}}$$

$$\langle W_{ij,l}^2 \rangle = \frac{e^{h_{ij,l}^{(n)}} + e^{-h_{ij,l}^{(n)}}}{e^{h_{ij,l}^{(n)}} + e^{-h_{ij,l}^{(n)}} + e^{-g_{ij,l}^{(n)}}} .$$

In the backward pass, we substitute Eq. D.4 into Eq. 4.5, obtaining

$$h_{ij,l}^{(n)} = \frac{1}{2} \ln \frac{P\left(W_{ij,l} = 1 | D_n\right)}{P\left(W_{ij,l} = -1 | D_n\right)}$$

$$= h_{ij,l}^{(n-1)} + \frac{1}{\sqrt{K_l}} \Delta_{i,l} \nu_{j,l-1} ,$$

as for the binary weights. Similarly,

$$g_{ij,l}^{(n)} = \frac{1}{2} \ln \left[ \frac{P\left(W_{ij,l} = 1 | D_n\right) P\left(W_{ij,l} = -1 | D_n\right)}{P\left(W_{ij,l} = 0 | D_n\right)} \right]$$

$$= g_{ij,l}^{(n-1)} + 0 .$$

Therefore, the parameter $g_{ij,l}$ is not updated. In the end we choose the MAP weight configuration (using Eq. 4.6)

$$W_{ij,l}^* = \mathcal{I}\left\{|h_{ij,l}| > g_{ij,l}\right\} \operatorname{sign}\left(h_{ij,l}\right) .$$

Therefore, the initial conditions on $g_{ij,l}$ (*i.e.*, the prior) act as a threshold which generates sparse weights.

## D.3  Real-valued weights

Suppose $S_{ij,l} = \mathbb{R}$, so $W_{ij,l}$ can receive any real value. A naive implementation of the algorithm would require an infinite number of updates, for each possible value of $W_{ij,l}$. A simple way to circumvent this is to as assume that each real weight can be written as an infinite sum of binary weights $W_{ij,l}^{\alpha}$

$$W_{ij,l} = \lim_{A \to \infty} \frac{1}{\sqrt{A}} \sum_{\alpha=1}^{A} W_{ij,l}^{\alpha} ,$$

where each binary weight is parametrized as in D.1 with parameter $h_{ij,l}^{\alpha(n)}$, and we denote

$$h_{ij,l}^{(n)} = \lim_{A \to \infty} \frac{1}{\sqrt{A}} \sum_{\alpha=1}^{A} h_{ij,l}^{\alpha(n)} .$$

Using Eq. D.2, we obtain for each binary weight

$$h_{ij,l}^{\alpha(n)} = h_{ij,l}^{\alpha(n-1)} + \frac{1}{\sqrt{A}} \frac{1}{\sqrt{K_l}} \Delta_{i,l} \nu_{j,l-1}. \qquad (D.5)$$

This immediately gives

$$h_{ij,l}^{(n)} = h_{ij,l}^{(n-1)} + \frac{1}{\sqrt{K_l}} \Delta_{i,l} \nu_{j,l-1}.$$

Assuming $h_{ij,l}^{\alpha(0)} \propto 1/\sqrt{A}$, from Eq. D.5 we also have $h_{ij,l}^{\alpha(n)} \propto 1/\sqrt{A}$. Therefore, after step $n$,

$$\langle W_{ij,l} \rangle = \lim_{A \to \infty} \frac{1}{\sqrt{A}} \sum_{\alpha=1}^{A} \tanh\left(h_{ij,l}^{\alpha(n)}\right)$$

$$= \lim_{A \to \infty} \frac{1}{\sqrt{A}} \sum_{\alpha=1}^{A} h_{ij,l}^{\alpha(n)} = h_{ij,l}^{(n)}$$

and

$$\mathrm{Var}\left(W_{ij,l}\right) \quad = \quad \lim_{A\to\infty} \frac{1}{A} \sum_{\alpha=1}^{A} \left[1 - \tanh^2\left(h_{ij,l}^{\alpha(n)}\right)\right] = 1$$

Therefore, using CLT, we have after the $n$-th update

$$W_{ij,l} = \lim_{A\to\infty} \frac{1}{\sqrt{A}} \sum_{\alpha=1}^{A} W_{ij,l}^{\alpha} \sim \mathcal{N}\left(h_{ij,l}^{(n)}, 1\right)$$

And so, our MAP estimate of $W_{ij,l}$ is

$$W_{ij,l}^{*} = \mathrm{argmax}_{W_{ij,l}} P\left(W_{ij,l}|D_n\right) = h_{ij,l}^{(n)}. \tag{D.6}$$

# E    Numerical Experiments - additional details

## E.1    Implementation

We tested the EBP algorithm in two special cases - a binary MNN (Algorithm 1) and a real MNN (Algorithm 2). In the first, after each update step, the MAP configuration for all the binary weights, is given by taking the sign of $h_{ij,l}$ (Eq. D.3), and the value of $h_{i0,l}$ is used for the biases (Eq. D.6). The EBP-D output is then obtained by substituting the MAP configuration into Eq. 2.2 and obtaining $y = v_L$. The EBP-P output is calculated using $y = \mathrm{sign}\left(\langle v_L \rangle\right)$. The second MNN has real weights, and is trained by Algorithm 2. The MAP configuration is given by $h_{ij,l}$ itself for all the synaptic weights and biases (Eq. D.6). Again, in the EBP-D $y = v_L$ and in EBP-P the output is calculated using $y = \mathrm{sign}\left(\langle v_L \rangle\right)$.

All algorithms were run using Matlab 2013b. Note that EBP on real MNNs (Algorithm 2) is very similar to BP, and hence should have similar running times - as was observed in practice. EBP on binary MNNs (Algorithm 1) performs additional non-linear operations on the weights ($\tanh\left(\cdot\right)$, or $\mathrm{sech}\left(\cdot\right)$), and therefore is expected to be somewhat slower. In practice, it was two times slower than BP if the values of the non-linear operation were saved and re-used, or five times slower if these values were not saved (to reduce memory requirements).

In all data sets we centralized (removed the means) and normalized the input (so $\mathrm{std} = 1$), as recommended for BP [22]. Both in BP and EBP algorithms we used uniform initial conditions, with std=1, as recommended for BP [22], so $\sqrt{K_l/3}h_{ij,l}^{(0)} \sim \mathrm{U}\left[-1,1\right]$ for EBP, and similarly for $W_{ij,l}^{(0)}$ in BP. In BP we used an activation function $f\left(x\right) = 1.7159 \tanh\left(2x/3\right)$ for the hidden neurons, as recommended by [22]. If cross-entropy (CE) loss was used, then in the output neurons we used the relevant logistic activation functions $f\left(x\right) = \left(1 + e^{-x}\right)^{-1}$ [6]. Parameter scans for the learning rate in BP was performed over

$$\left\{10^{-4}, 3\cdot 10^{-4}, 5\cdot 10^{-4}, 8\cdot 10^{-4}, 10^{-3}, 3\cdot 10^{-3},\right.$$
$$\left. 5\cdot 10^{-3}, 8\cdot 10^{-3}, 10^{-2}, 3\cdot 10^{-2}, 5\cdot 10^{-2}, 8\cdot 10^{-2}, 0.1\right\}$$

and in AROW we scanned the regularization parameter over $\left\{10^k\right\}_{k=-4}^{4}$ (the results were rather insensitive to changes in that parameter). The optimal parameters (which yield the best performance), given in Fig. E.1 are never near the edges of the scanned range.

## E.2    Additional results

**Small fan-in.**    To check whether the algorithm can work if the large fan-in assumption is incorrect, we also performed small-scale classification using the Pima Indians Diabetes dataset [2]. The set contains 768 instances with 8 features and 2 classes. The task was to identify the $\mathrm{label} \in \{-1, +1\}$, using a $8 \to 200 \to 1$ MNN classifier. Classification error was calculated using 10-fold cross validation, so we can compare with the previous best results reported in [1]. Results are shown in Table 2: as can be seen, EBP-P still exhibits the best performance with binary weights, and the second best performance with real weights.

**Algorithm 1** A single update step of the the Expectation BackPropagation (EBP) algorithm for fully connected binary MNNs - with binary synaptic weights and real bias. We denote $\nu_{k,l} = \langle v_{k,l} \rangle$, $\tanh(h_{ij,l}) = \langle W_{ij,l} \rangle$, and $\mathcal{H}$ as the set of all $h_{ij,l}$.

**Function** $[\boldsymbol{\nu}_L, \mathcal{H}_{\text{next}}] = \text{UpdateStepBinaryMNN}(\mathbf{x}, \mathbf{y}, \mathcal{H})$

% **Forward pass**
Initialize
$$\forall k : \nu_{k,0} = x_k, \forall l : \nu_{0,l} = 1$$

**for** $m = 1$ **to** $L$ **do**
  $\forall k:$

$$\mu_{k,m} = \frac{1}{\sqrt{K_{m-1}}} \left[ h_{k0,m} + \sum_{r=1}^{V_{m-1}} \tanh(h_{kr,m}) \nu_{r,m-1} \right]$$

$$\sigma_{k,m}^2 = \frac{1}{K_{m-1}} \left[ 1 + \sum_{r=1}^{V_{m-1}} \left[ \left(1 - \nu_{r,m-1}^2\right)\left(1 - \delta_{1m}\right) + \nu_{r,m-1}^2 \operatorname{sech}^2(h_{kr,m}) \right] \right]$$

$$\nu_{k,m} = 2\Phi\left(\mu_{k,m}/\sigma_{k,m}\right) - 1$$

**end for**
% **Backward pass**
Initialize
$$\Delta_{i,L} = y_i \frac{\mathcal{N}\left(0 | \mu_{i,L}, \sigma_{i,L}^2\right)}{\Phi\left(y_i \mu_{i,L}/\sigma_{i,L}\right)}$$

**for** $l = L$ **to** $1$ **do**

$$\forall i : \Delta_{i,l-1} = \frac{2}{\sqrt{K_{l-1}}} \mathcal{N}\left(0 | \mu_{i,l-1}, \sigma_{i,l-1}^2\right) \sum_{j=1}^{V_m} \tanh(h_{ji,l}) \Delta_{j,l}$$

$$\forall i,j : h_{ij,l}^{\text{next}} = h_{ij,l} + \frac{1}{\sqrt{K_{l-1}}} \Delta_{i,l} \nu_{j,l-1}$$

**end for**

Figure E.1: Optimal parameter values for **(A)** BP and **(B)** AROW. Red lines - Minimum and maximal values of the scan.

**Algorithm 2** A single update step of the algorithm for a fully connected MNNs with real weights and bias. We denote $\nu_{k,l} = \langle v_{k,l} \rangle$, $h_{ij,l} = \langle W_{ij,l} \rangle$, and $\mathcal{H}$ as the set of all $h_{ij,l}$.

**Function** $[\boldsymbol{\nu}_L, \mathcal{H}_{\text{next}}] = \text{UpdateStepReaMNN}(\mathbf{x}, \mathbf{y}, \mathcal{H})$

% **Forward pass**

Initialize
$$\forall k : \nu_{k,0} = x_k, \forall l : \nu_{0,l} = 1$$

**for** $m = 1$ **to** $L$ **do**
    $\forall k$:

$$\mu_{k,m} = \frac{1}{\sqrt{K_{l-1}}} \sum_{r=0}^{V_{m-1}} h_{kr,m} \nu_{r,m-1}$$

$$\sigma_{k,m}^2 = 1 + \frac{1}{K_{l-1}} \sum_{r=0}^{V_{m-1}} \left[ \left( \nu_{r,m-1}^2 - 1 \right) \delta_{m1} + (1 - \delta_{m1}) \left( 1 - \nu_{r,m-1}^2 \right) h_{kr,m}^2 \right]$$

$$\nu_{k,m} = 2\Phi \left( \mu_{k,m} / \sigma_{k,m} \right) - 1$$

**end for**

% **Backward pass**

Initialize
$$\Delta_{i,L} = y_i \frac{\mathcal{N} \left( 0 | \mu_{i,L}, \sigma_{i,L}^2 \right)}{\Phi \left( y_i \mu_{i,L} / \sigma_{i,L} \right)}$$

**for** $l = L$ **to** $1$ **do**

$$\forall i : \Delta_{i,l-1} = \frac{2}{\sqrt{K_{l-1}}} \mathcal{N} \left( 0 | \mu_{i,l-1}, \sigma_{i,l-1}^2 \right) \sum_{j=1}^{V_m} h_{ji,l} \Delta_{j,l}$$

$$\forall i,j : h_{ij,l}^{\text{next}} = h_{ij,l} + \frac{1}{\sqrt{K_{l-1}}} \Delta_{i,l} \nu_{j,l-1}$$

**end for**

| Dataset | Previous Best[1] | Real EBP-D | Real EBP-P | Binary EBP-D | Binary EBP-P | BP | Clipped BP |
|---|---|---|---|---|---|---|---|
| Pima Indians diabetes | 22.3% | 23.82% | 22.11% | 26.18% | **21.6%** | 22.9% | 34.9% |

Table 2: Pima Indians Diabetes dataset - Test error.

**Deeper architectures.** To verify that the MNN's depth does not effect our conclusions, we test our algorithms using the same setup (except for more training epochs - eight instead of three) on a a deeper architecture of $M \rightarrow 1000 \rightarrow 100 \rightarrow 1$. The MNN was tested on three of the datasets. Results are described in Table 3. As can be seen, for these datasets EBP-P exhibits the best performance with binary weights, and the second best performance with real weights. This is the same as in the 2-layer case, except for the Reuters news I8 dataset - where before EBP performed better with real weights than with binary weights.

| Dataset | Real EBP-D | Real EBP-P | Binary EBP-D | Binary EBP-P | BP | Clipped BP |
|---|---|---|---|---|---|---|
| Reuters news I8 | 15.7% | 15.5% | 21.5% | **15.25%** | 17.2% | 25.4% |
| 20News group comp vs HW | 5.06% | 5.01% | 6.2% | **4.39%** | 8.26% | 12.75% |
| Spam or ham d0 | 1.12% | 0.88% | 3.08% | **0.72%** | 1.92% | 10.54% |

Table 3: Test error for a 3-layer MNN.

# F Comparison with Backpropagation

The EBP algorithm for MNNs with real weights (summarized in Algorithm 2) is almost identical to the standard BP algorithm, when the variables $h_{ij,l}$ are interprets as the real-valued weights in a BP algorithm. To see this, recall [22] that in BP we wish to train a MNN of the form

$$\mathbf{u}_l = \mathbf{W}_l \mathbf{v}_{l-1}$$
$$\mathbf{v}_l = f(\mathbf{u}_l),$$

$\forall l = 1, \ldots, L$, where $f(\cdot)$ is some sigmoid function. The training is done by minimizing an error function $E(\mathbf{y}, \mathbf{v}_L)$, through the following recursive equations. First, we initialize

$$\Delta_{i,L} = - \eta \frac{\partial E(\mathbf{y}, f(\mathbf{u}_l))}{\partial u_{i,L,}}, \tag{F.1}$$

where $E$ is some non-negative error function and $\eta$ is a learning rate. Then, for $l = L, \ldots, 1$ and $\forall i, j$ we calculate

$$\Delta_{i,l-1} = f'(u_{i,l-1}) \sum_{j=1}^{V_m} W_{ji,l}^{(n-1)} \Delta_{j,l}. \tag{F.2}$$

$$W_{ij,l}^{(n)} = W_{ij,l}^{(n-1)} + \Delta_{i,l} v_{j,l-1}. \tag{F.3}$$

where $f'$ is the derivative of $f$ Comparing with EBP for a MNN with real-valued weights (summarized in Algorithm 2), we find that it is nearly identical. Specifically, in BP, we just need to substitute $W_{ij,l} = h_{ij,l}/\sqrt{K_l}$, $v_{k,m} = \langle v_{k,m} \rangle$, use $\eta = 1$, the activation function

$$f(u_{k,l}) = 2\Phi(u_{k,l}/\sigma_{i,l}) - 1$$

and the "cross-entropy" [6] error function

$$E(\mathbf{y}, \mathbf{v}_L) = -\ln P(\mathbf{v}_L = \mathbf{y}) = -\sum_i \ln \left( \frac{1 + y_i v_{i,L}}{2} \right).$$

The only difference is that the input $u$ to each neuron is scaled adaptively through $\sigma_{i,l}$ - which depends on the inputs and weights (Eq. 4.2). This implies that the EBP algorithm is invariant to changes in the amplitude of of the input $\mathbf{x}$ (*i.e.*, $\mathbf{x} \rightarrow c\mathbf{x}$, where $c > 0$). This preserves the amplitude invariance of the sign activation function we used in the original MNN (Eq. 2.2). Note that in the standard BP algorithm the performance is directly affected by the amplitude of the input, so it is a recommended practice to re-scale it in pre-processing [22]. Interestingly, is also recommended practice to use the cross-entropy error function for classification tasks [6], and to scale initial conditions $W_{ij,l}^{(0)} \sim 1/\sqrt{K_l}$ [22]. These rather heuristic practices naturally arise in EBP, which was derived from first principles.

Figure G.1: Approximating EBP-P output by averaging the output of a random sample of MNNs with binary weights (#samples=$\left\{2^k\right\}_{k=0}^9$ shown). *Top figure:* $P\left(y_{\text{EBP}-\text{P}} \neq y_{\text{sampling}}\right)$ - sign error between the sampling output and analytical output. We show the analytically predicted error (Eq. G.2), as well as the empirical error from the sampling simulation. *Bottom figure:* The test classification error of the sampling output $y_{\text{sampling}}$ in comparison to the same error of the analytical $y_{\text{EBP}-\text{P}}$ output. Note that already for 16 samples (the fifth point), we get a comparable error. Error bars give the $95\%$ confidence intervals.

## G   Sampling the weights

The EBP-P output (Eq. 4.7) is the MAP estimate of the MNN output (Eq. 3.2) which typically gives the best performance empirically (Table 1). In the paper we calculated the EBP-P output analytically (Eq. 4.7). Motivated by hardware applications, we would like, instead, to calculate the EBP-P output by averaging the output of several MNNs with binary weights. In this section we explain how this is done, and calculate analytically and numerically the approximation error for such MNNs with a single output neuron (*i.e.*, $V_L = 1$ - a binary classification task).

Originally the EBP-P output was derived from an ensemble average over the output of all such MNNs when the weights are distributed according to the posterior

$$P\left(\mathcal{W}|D_N\right) = \prod_{i,j,l} P\left(W_{ij,l}|D_N\right) \tag{G.1}$$

where $P\left(W_{ij,l}|D_N\right)$ is the weight distribution given by the algorithm (section 4) after training. In the binary output case, the classification according to the EBP-P output was peformed according to (Eq. 4.7)

$$\begin{aligned} y_{\text{EBP}-\text{P}} &= \text{sign}\left(\langle v_L \rangle\right) \\ &= \text{sign}\left(\mu_L\right), \end{aligned}$$

where either $\langle v_L \rangle$ or $\mu_L$ can calculated analytically (Eqs. 4.1-4.2) from the distribution over $\mathcal{W}$ (Eq. G.1). However, in order to implement this output in hardware using binary MNNs, we will use instead the following sampling-based procedure.

First, we generate $S$ samples of the weights $\left\{\mathcal{W}^{(s)}\right\}_{s=1}^{S}$ by sampling from the inferred distribution (Eq. G.1). Then, we use each sample $\mathcal{W}^{(s)}$ to calculate the input to the last layer in the MNN (Eq. 2.2) for each example $\mathbf{x}$ in the test set

$$\forall s: \ u_L^{(s)} = \mathbf{W}_L^{(s)}\text{sign}\left(\mathbf{W}_{L-1}^{(s)}\text{sign}\left(\cdots\mathbf{W}_1^{(s)}\mathbf{x}\right)\right) \ .$$

Then, we perform the classification according to

$$y_{\text{sampling}} = \text{sign}\left(\frac{1}{S}\sum_{s=1}^{N}u_L^{(s)}\right).$$

Due to the CLT approximation (Eq. 3.14), we have

$$\frac{1}{S}\sum_{s=1}^{N}u_L^{(s,n)} \sim \mathcal{N}\left(\mu_L, \frac{\sigma_L^2}{S}\right).$$

Therefore, it is straightforward to calculate the following error probability

$$
\begin{aligned}
P\left(y_{\text{EBP}-\text{P}} \neq y_{\text{sampling}}\right) &= P\left(\text{sign}\left(\frac{1}{S}\sum_{s=1}^{N}u_L^{(s)}\right) \neq \text{sign}\left(\mu_L\right)\right) \\
&= \Phi\left(-\sqrt{S}\left|\frac{\mu_L}{\sigma_L}\right|\right) \\
&= \Phi\left(-\sqrt{S}\left|\Phi^{-1}\left(\frac{\langle v_L\rangle + 1}{2}\right)\right|\right) .
\end{aligned}
\tag{G.2}
$$

Therefore, asymptotically, the error will decay exponentially fast in $S$, since

$$\lim_{x\to\infty}\Phi\left(-x\right) \sim \frac{1}{x\sqrt{2\pi}}e^{-x^2/2}.$$

Next, we examine numerically this convergence speed, using the Spam or ham d0 dataset. Importantly, the above sampling-based method only assumes that the CLT theorem can be used to approximate the input to each neuronal layer. This is only approximately true for finite fan-in $K$. Empirically, we noticed that CLT was usually accurate - except in the input layer for a few examples. In these cases, the inputs were "heavy tailed" - so a single feature (an input component) was much stronger then all the others[8]. On these examples, the sampling output $y_{\text{sampling}}$ might not converge to $y_{\text{EBP}-\text{P}}$, even if $S \to \infty$. To correct for this, we split any "strong" features which cause this issue (before any pre-processing). Specifically on this dataset (Spam or ham d0), we select any feature which contained examples deviating more than 140 standard deviations from the mean . Then we split that feature into five identical features with their original value divided by five. In total, this results in a modest $26\%$ increase in the number of features (*i.e.*, the size of the input layer). As can be seen on Figure G.1, the sampled output quickly converges to the anaytical output of EBP-P.

## Footnotes

[1]*i.e.*, having more than a single layer of adjustable weights.

[2]*i.e.*, having only a single layer of adjustable weights.

[3]MNN with stochastic activation functions will have a "smoothed out" version of this.

[4]Due to numerical inaccuracy, calculating $\Delta_{i,L}$ using Eq. 4.3 can generate nonsensical values ($\pm\infty$, NaN) if $|\mu_{i,L}/\sigma_{i,L}|$ becomes to large. If this happens, we use instead the asymptotic form in that limit

[5]This approximation is not required if all neurons in the MNN have a fan-out of one.

[6]Which departs from the online framework considered here, since it requires two samples in each update.

[7]*i.e.*, activation functions for which $\left(f\left(\mathbf{x}\right)\right)_i \neq f\left(x_i\right)$, such as softmax or argmax.