[Reviews · NeurIPS 2014]

Submitted by Assigned_Reviewer_5

The authors propose an EP/TAP-like algorithm for learning a distribution on the
weights of an MLP. They derive variational learning rules for binary and
real-valued weights, based upon a CLT argument, and show that the binary valued
weights perform best on several data sets.

The setup and motivation is appealing and well done. I'm surprised this has not
been done before! The experimental results combined with the apparent simplicity
are particularly encouraging.

The paper is largely well-written, but I found the notation awkward and
hard to follow. For example, I understand rbracket W_ij,l lbracket to be the
mean of W_kr,m, but I do not see it defined anywhere. Thus I cannot see how to
implement the proposed algorithm. The switch between mu(W_ij,l) and rbracket
W_ij,l lbracket is the source of this confusion.

A few comments:

I believe the authors ran AROW themselves: was it their own implementation? Do they consider the results to be consistent with those in the
original AROW paper?

Considering (4.6), since (4.5) appears to be linear in W_ij,l, does the max
just depends upon the sign of the coefficient of W_ij,l?

The proposed method scaled quadratically in the number of weights. Graves
(2011) proposes a variational learning algorithm that scales linearly; this is
worth citing, at least.
Summary: A variational method for inferring a posterior over the weights of a neural network, based upon a CLT argument. Positive results on several data sets compared to AROW.

Submitted by Assigned_Reviewer_15

This paper proposes an approximation to Bayesian learning of multi-layer neural networks. They use a mean-field approximation to the distribution over weight matrices, and over unit inputs, plus a couple approximations based on the assumption that each unit has a large number of inputs. Given the resulting approximation to the posterior distribution over weights, they can marginalise over weights to compute the best output. They apply this framework to a variety of NN architectures. The resulting algorithm is similar to standard back-propagation, but without the need to set meta-parameters. Empirical evaluation on some text classification benchmarks achieve state of the art results.

This is a well motivated framework with interesting resulting models and impressive empirical results. It gives a principled Bayesian approach to learning in multi-layer NNs, including removing the need for the many hacks to set meta parameters like the learning rate schedule. The success of this variational approximation opens the way for alternative underlying models, and alternative approximations.

The main body of the paper is not entirely understandable without also reading the annexes, which are longer than the paper itself. This starts in section 3.4, but more worryingly it continues in section 4, where the algorithm is actually specified. It wouldn't take much space to provide some more intuitive explanations of the variables and the equations by linking them to the previous discussion. As it stands, section 4 is a bit too much like a core dump.

The empirical comparison to BackPropagation is not a fair one. I don't think that doing only 3 iterations over ~2000 examples with a 1st order gradient descent method, and constant learning rate, can be considered state of the art.

The experiments are all on text classification problems, which have large sparse input spaces (bag of words), and presence/absence of features (words) is more important than their real values (frequencies). This may be why (distributions over) binary weights and a threshold activation function work well here.

Summary: This is a well motivated application of Bayesian parameter estimation to multi-layer neural networks. The proposed approximation is efficient and effective.

Submitted by Assigned_Reviewer_34

The paper introduces a back propagation algorithm which instead of computing the gradient information, updates beliefs regarding posterior distributions of model parameters. In doing so the learning algorithm relies on a mean-field approximation + the gaussian assumption (with the diagonal covariance matrix). They call this approach “expectation backdrop” (EBP).

Overall, I find some aspects of this work quite interesting. My questions / concerns / comments:

1) The paper needs to provide a better discussion / comparison to the previous work on Bayesian learning of NNs ( see e.g., Radford Neal’s and David MacKay’s work from mid-90s + more recent work, e.g., on Langevin dynamics, ...). There have been a lot of work, and almost nothing is discussed here.

2) The introduction of the EBP algorithm is motivated by arguing that the standard BP algorithm is not capable of learning NNs with binary weights which, according to the authors, much more efficient from the computational perspective at test time (however, one would need larger networks then?). The EBP method is indeed capable of learning binary NNs in a fairly effective way. The catch is though that using binary models directly (i.e. the MAP solution) at test time directly does not seem to lead to competitive results (see column “Binary EBP-D” in Table 1: the error rate is >= doubled for 4 datasets out of 7). What seems to work well is Bayesian model averaging with the binary model (“Binary EBP-P”) but this is again a ‘continuous’ computation which is at least as expensive as the computation with a normal NN. At the very least the authors need to clarify this point, currently, as the binary version is the key motivation of this research (see the first paragraph of the paper).

3) The notation in the model section needs to be cleaned up a bit. E.g., I was slightly confused with v vs. \nu (which is actually never formally introduced). The appendix was helpful but the paper needs to be understandable on its own.

4) It would be interesting to see (empirically) how well this approach generalizes to deeper architecture (only models with a single hidden layers are considered in the paper).

5) Since the argument is efficiently, I’d like to see a bit more discussion of this aspect in the paper (in the experimental section, e.g.). The approach is much more complex than standard BP (from the implementation standpoint), and results even of the EBP-P version are not so obviously better than the ones for BP, so it would be nice to have more details to convince the readers. (However, the appendix does include some asymptotic analyses).
Summary: Though some ideas in the paper are interesting, I have some questions regarding evaluation, discussion of related work. The paper is not terribly well written.
Author Feedback
Author rebuttal: We thank the reviewers for their valuable and constructive input. Most of the raised issues are the result of our effort to make the paper short, readable and focused – which, unfortunately, sometimes resulted in confusing omissions and over-simplifications. However, all these can (and will) be easily corrected.

*Rev. 15:
- Agreed - we will improve sections 3.4 and 4.
- In all the algorithms (including BP) in section 5, we did not observe performance improvements after 3 iterations (we will add learning curves to show this), so we did not bother to use more. Comparison with BP+"tricks" (e.g., weight decay or momentum) is important, but more involved. For example, we derived EBP assuming an online setting (only one sample at a time), while momentum uses information from two samples. A fair comparison would require us to derive EBP given two samples (which is possible, but beyond the scope of this paper). For such reasons, in this paper we only give “vanilla” BP as a “sanity check” and focus more on our comparison with AROW which is indeed state-of-the-art (ref [6]) for this task. We will clarify this point.
- A very interesting insight. We will add this, as we were not sure why the binary weights work better than real weights here.

*Rev. 34:
We agree with all of the main comments.
1) For brevity, we focused only on the Bayesian NNs literature which dealt with discrete weights (refs [1,5,18,20]). In retrospect, we agree that the more general context should have been clarified. Briefly, the works by Neal and MacKay, implemented for real weights, differ from ours also in several other respects: (1) their NNs have built-in noise (while our NNs are deterministic), which requires hyper-parameters; (2) their main aims are to determine hyper-parameters and perform model averaging; and (3) they use MCMC techniques - which are rather slow. Still today, Bayesian neural nets are difficult to scale to very large network sizes. However, on small datasets, Bayesian NNs can outperform state-of-the-art NNs. These issues were recently explained in “Dropout: A Simple Way to Prevent Neural Networks from Overfitting” (section 6.4 + Table 8), by Hinton and colleagues. We believe our method is an efficient approximation, which may be useful for accelerating Bayesian NNs training. We are not aware of the recent Langevin dynamics works (any reference will be appreciated).
2) Indeed, this was not clear. As we mentioned in the paper, the EBP-P output can be implemented by averaging the output of binary NNs when the weights are sampled from the inferred posterior. But the reviewer is right – we did not quantify how many samples are needed for this (as there are a few non-trivial issues here – but we can expand more about this in the supplementary material). The bottom line is that we calculated analytically and numerically the error of the output of this ensemble average, and that it decays fast with the number of samples. Asymptotically, it decays exponentially. Empirically, we needed to average between 4 to 64 samples to obtain a test error similar to EBP-P. Such a 4-64 effective increase in size (the number of NNs we need to average on) is modest in comparison to the orders of magnitude increases in speed (~10^3) and energetic efficiency (~10^5) expected to result from a binary implementation (ref. [11]).
3) Agreed. Specifically, though nu is defined in Eq. 4.1, there were a few confusing typos, and we accidently omitted the explanation that nu is the mean of v.
4) We performed additional simulations with deeper 3-layer architectures on these text datasets. Binary EBP-P performs the similarly, and also the ranking between the algorithms remains the same (performance: EBP-P binary >EBP-P real > AROW > BP). We are not sure if this is because of the domain (deep architectures might be less effective for text), or because we did not use additional “tricks” (e.g. convolutional architecture, or unsupervised pre-training) which are very useful for such deep networks.
5) Agreed, this discussion should be extended in the paper. However, we are somewhat confused about two comments. First, we are not sure why the implementation is more complex than BP. Just to clarify, the resulting algorithm is almost identical to BP (with the same efficiency) in the real case (appendix F), and very similar in other cases. Therefore, though the derivation of the algorithm is more complex than BP, the end result is not more complex. Second, it is not clear to us why, according to the reviewer, the results of EBP-P are not obviously better than the ones for BP – note we achieve about 40%-100% error decrease in 7 out of 8 benchmarks.

*Rev. 5:
Notation: Yes, rbracket W_ij,l lbracket is the mean of W_ij,l (not W_kr,m), which can be directly found from the posterior of W_ij,l (as in appendix D.1). Though the bracket notation was defined in line 89, we will add a reminder before it is used (We are not sure exactly which switch is confusing - is it Eq. 4.1?). We will publish the algorithm’s code on our website.
- We did run AROW ourselves, using the original code from ref [6]. There is only one (regularization) parameter to tune, so we performed a scan (Fig E.1B). The performance was not very sensitive to the value of the parameter. All AROW results we report are similar (or better) than in ref [6].
- Regarding 4.5-4.6: indeed, in the binary case, the max just depends on the sign of the cumulative sum of the coefficients of W_ij,l (see Eq. D.2-3). However, this is not generally true (e.g., real case - see D.6).
- The final algorithm is actually linear in the number of weights (we mention this on line 73), similarly to BP (to which it is almost identical). In case this was the cause of the confusion, note the quadratic complexity on line 601 was before we used the BP-like derivation (in the Appendix C) to improve further. We will clarify this, and add the Graves reference.